# Codesigning a systemic discharge intervention for inpatient mental health settings (MINDS): a protocol for integrating realist evaluation and an engineering-based systems approach

Corinna Hackmann [1,2] Alexander Komashie,[3] Melanie Handley,[4] Jamie Murdoch,[5] Adam P Wagner [6,7] Lisa Marie Grünwald [1] Sam Waller,[8] Emma Kaminskiy,[9] Hannah Zeilig,[10] Julia Jones,[11] Joy Bray,[12] Sophie Bagge,[13] Alan Simpson,[14] Sonia Michelle Dalkin [15] John Clarkson,[8] Giovanni Borghini,[16] Timoleon Kipouros [17] Frank Rohricht,[18,19] Zohra Taousi,[16] Catherine Haighton,[15] Sarah Rae,[12] Jon Wilson[1,2]

SR and JW are joint first authors.

For numbered affiliations see end of article.

**Correspondence to**
Dr Corinna Hackmann;
Corinna.hackmann@nsft.nhs.uk

## ABSTRACT

**Introduction** Transition following discharge from mental health hospital is high risk in terms of relapse, readmission and suicide. Discharge planning supports transition and reduces risk. It is a complex activity involving interacting systemic elements. The codesigning a systemic discharge intervention for inpatient mental health settings (MINDS) study aims to improve the process for people being discharged, their carers/supporters and staff who work in mental health services, by understanding, co-designing and evaluating implementation of a systemic approach to discharge planning.

**Methods and analysis** The MINDS study integrates realist research and an engineering-informed systems approach across three stages. Stage 1 applies realist review and evaluation using a systems approach to develop programme theories of discharge planning. Stage 2 uses an Engineering Better Care framework to codesign a novel systemic discharge intervention, which will be subjected to process and economic evaluation in stage 3. The programme theories and resulting care planning approach will be refined throughout the study ready for a future clinical trial. MINDS is co-led by an expert by experience, with researchers with lived experience co-leading each stage.

**Ethics and dissemination** MINDS stage 1 has received ethical approval from Yorkshire & The Humber—Bradford Leeds (Research Ethics Committee 22/YH/0122). Findings from MINDS will be disseminated via high-impact journal publications and conference presentations, including those with service user and mental health professional audiences. We will establish routes to engage with public and service user communities and National Health Service professionals including blogs, podcasts and short videos.

**Trial registration number** MINDS is funded by the National Institute of Health Research (NIHR 133013) https://fundingawards.nihr.ac.uk/award/NIHR133013. The realist review protocol is registered on PROSPERO.

**PROSPERO registration number** CRD42021293255.

## STRENGTHS AND LIMITATIONS OF THIS STUDY

⇒ Codesigning a systemic discharge intervention for inpatient mental health settings (MINDS) was conceived from lived experience and is aiming for high standards of coproduction.

⇒ MINDS incorporates realist methods with a systems-based engineering approach to understand and improve discharge experience and outcomes.

⇒ The complexity and pace of delivery and funding constraints might be barriers to coproduction, but the team has extensive experience, is evaluating levels of coproduction and will report these.

⇒ Some aspects of the MINDS project, including the translation of lived experience into outputs, are flexible and responsive to emerging discoveries; consequently, there may be refinements to the protocol—these will be outlined in related MINDS publications.

## INTRODUCTION

The transition period following discharge is high risk; around 13% of people are quickly readmitted[1] and rates of suicide have been found to be 191 times higher for working age adults compared with matched-age comparators.[2] A wide range of factors contribute to relapse following discharge, including feeling overwhelmed, managing mental health symptoms, returning to roles and day-to-day pressures of life.[3] Discharge planning supports transition and reduces risk by identifying postdischarge needs and how to manage these.[3 4] National Institute of Health and Care Excellence guidance and the Care Quality Commission identify that discharge planning should be collaborative and person

centred[5 6] but provide limited clarity on how this should be achieved.

Evidence suggests that discharge is often inadequately planned with little involvement from the person being discharged and their carers/supporters, resulting in poor transition and increased risk.[7–10] The charity Mind surveyed 1221 people who had experienced discharge finding that 33% were given either no or insufficient notice of discharge, and for 37%, there was no plan post discharge.[11]

Discharge planning is complex and involves many multi-faceted interacting systemic elements. People are heterogeneous in terms of needs. Mental health service delivery is reliant on the reasoning, reactions and actions of staff, which are influenced by the wider system. Discharge planning, therefore, needs to address the needs of the person being discharged while working within organisational complexity, constraints and priorities.

Previous interventions neglect complex systemic factors that either support or undermine discharge planning.[12] Neglecting the wider system, including systemic pressures and the needs of staff, is likely to explain why previous attempts to improve discharge have failed. To establish effective discharge planning procedures, it is critical to understand the ward as a complex system (operating within wider systems and national policy). Acknowledging such complexity reframes health services research 'from investigations of complex social interventions to interventions in complex social systems',[13] echoing perspectives articulated across public health and global health systems research.[14 15] Despite growing recognition of the need for systems approaches in healthcare,[16–19] to date, no research has used this to improve mental health discharge.

## METHODS AND ANALYSIS
### Study design
The Codesigning a systemic discharge intervention for inpatient mental health settings (MINDS) study commenced 1 January 2022 and ends 1 January 2025. The MINDS study will innovatively integrate realist research and an engineering-informed systems approach to codesign a systemic approach to discharge. Realist reviews[20] and evaluations[21] can be foundational for complex intervention development as they explain how and why change occurs through causal mechanisms.[22] Discharge planning is a multifaceted activity involving many systemic elements including service users, healthcare staff, policy, documentation, information systems and external bodies. We will use a healthcare-based systems approach as a framework for building and refining programme theories of discharge planning that set out relationships of components across system levels in a sociotechnical context (ie, people interacting with each other and technical components such as electronic records). This will comprise multiple context–mechanism–outcome configurations to explain what works, for whom and in what circumstances. The programme theories will inform system design solutions for discharge (see below for an example).

Engineering Better Care is an established engineering systems approach for the design of safe and successful healthcare delivery.[23–25] The framework involves four key perspectives[26]: (1) 'People' focuses on the needs of key stakeholders; (2) 'Systems' explores the interactions between stakeholders and layers of the system; (3) 'Design' encourages innovation and investigates issues before proposing solutions; (4) 'Risk' predicts and models the risks associated with all proposed solutions. This framework will inform the codesign of a systemic approach to discharge. Implementation and economic implications will be evaluated to refine the discharge process.

Across the stages of the study, we will build evidence-based theories to codesign and trial a Systemic Discharge Approach that promotes collaborative discharge planning. For the purposes of this protocol, this will be referred to as 'the intervention'. It is anticipated that the intervention will be multifaceted, this may include training, changes to processes and documentation. It will be designed to address factors inhibiting collaborative discharge planning across the levels of the system with consideration to potential unintended consequences arising from new ways of working.

### Project aim
Co-produce and evaluate implementation and cost impact of a systemic approach to discharge.

### Project objectives
1. Understand discharge planning as a complex intervention.
2. Codesign a systemic discharge intervention.
3. Evaluate acceptability, implementation and cost–impact of the new discharge intervention.

### Research questions
1. How and in what contexts is mental health discharge currently performed?
2. Who are the primary stakeholders (eg, service users (The MINDS research team's preferred term for people who have lived experience of accessing mental services is 'people'. However, we have used the term 'service user' where discussing participants in the research project to delineate between different groups), carers/supporters, healthcare staff), how can they be characterised and what are their needs?
3. What are the successful outcomes for mental health discharge, how do these relate to contexts across the system and what are the mechanisms underlying this?
4. How can mental health discharge be improved?
5. How can this be implemented and measured?

### Patient and public involvement
The MINDS study is co-led by SR, who conceived the idea from lived experience of unhelpful discharge. Each stage is co-led by researchers with relevant lived experience. MINDS includes a Lived Experience Advisory Group (LEAG) comprised of people with relevant lived experience or experience of being a significant

carer or supporter. The LEAG will offer governance and support coproduction and key strategic decision-making throughout the project. The study steering committee (SSC) includes two members with lived experience, one of whom is cochair. The methodological approaches adopted align with current empirical healthcare research theory, including Medical Research Council complex intervention guidance.[27] Systems and realist approaches were selected as they also intrinsically value and prioritise key stakeholder perspectives and an iterative approach to knowledge generation.

## Case study sites

Sites were purposively sampled to represent geographically distinct (serving rural and urban communities) statutory mental healthcare organisations with different demographic profiles and mixed public inspection ratings. Three mental healthcare sites are included. Two wards will be selected from each site for ward observations and evaluation of the new intervention. MINDS recognises that minority ethnic service users are disproportionately detained under the UK Mental Health Act[28–30] and over-represented among psychiatric in-patients in UK statutory provision.[31] Consequently, we will monitor recruitment and employ targeted strategies to ensure the study sample reflects diversity of experience.

## Recruitment

Figure 1 details recruitment aims across the MINDS project. There will be diverse promotion of the study to ensure broad access to participation, including (but not limited to) posters in clinical areas, participation newsletters, attendance at participation events. Additionally, eligible individuals may be contacted by their clinical team, or where they have signed-up to be contacted for

research, by the research team to enquire whether they are interested in participation.

## Inclusion criteria
### Service users
Interviews, focus groups and workshops: all service users (18 years and over), accessing community mental health services in the case study sites, discharged within the previous 12 months (stage 1) or being discharged from a case study site ward (stage 3), will be eligible.

Ward-based observations: all service users (18 years and over) currently admitted on selected wards.

### Staff
Staff, working in participating mental healthcare organisations, whose role impacts (directly or indirectly) on inpatient discharge.

### Carers/supporters
Carers/supporters (people who identify as having a significant caring or supportive role) for people who have experienced inpatient discharge in one of the participating mental healthcare organisations within the last 12 months.

## RESEARCH PLAN
The MINDs' study operates across three stages.

### Stage 1: Understand discharge planning as a complex intervention
#### Aim
Build, test and refine evidence-based programme theories of discharge planning and preparation.

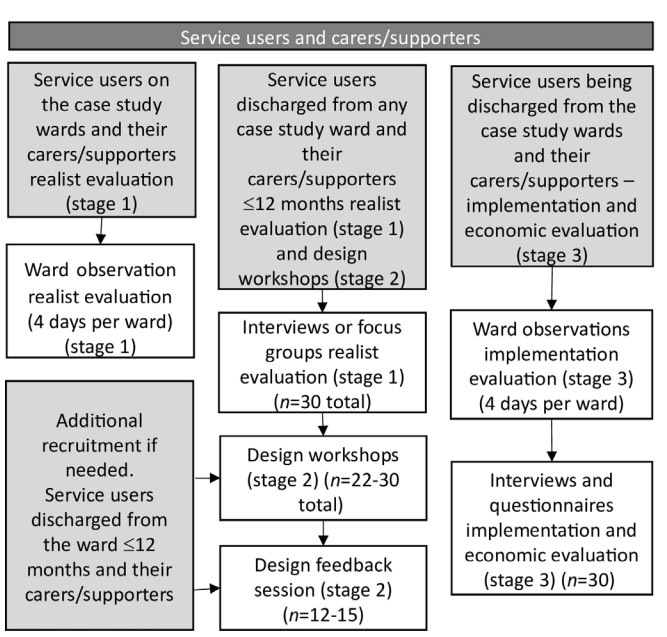

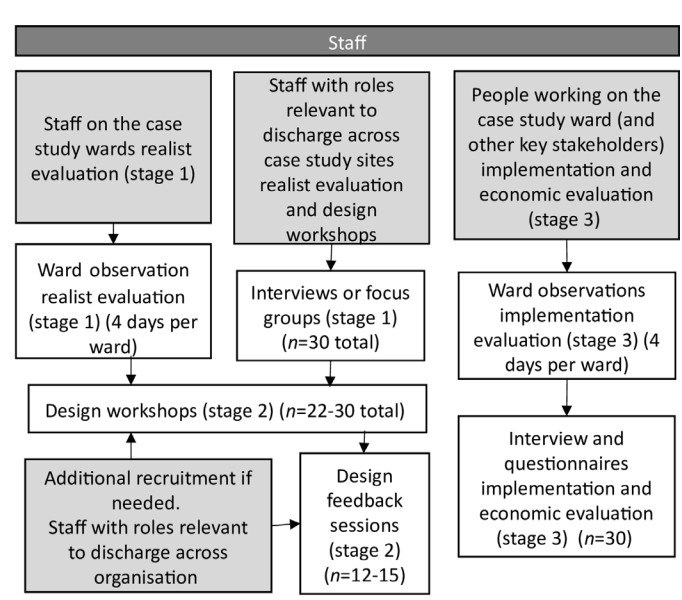

**Figure 1** Recruitment aims across the three stages.

### Objectives

1. Conduct a realist review integrating the Engineering Better Care systems approach to map and explain the relationship between key factors involved in discharge planning.
2. Identify service user needs for discharge planning.
3. Test programme theories in a realist evaluation across three case study sites.
4. Refine programme theories to inform codesign work in stage 2.

### Design

#### Realist review

A realist review will synthesise quantitative and qualitative evidence on service user, carer and staff experiences of, and interventions for, discharge planning. The review will result in evidence-based theories that include factors across all system levels to explain post-discharge outcomes. The review will consist of three iterative phases:

1. Defining review scope, concept mining and initial theory development: a series of meetings with the research team and LEAG will be used to define the system of interest. Initial programme theories will be developed from the literature identified from a systematic review[12] and an internal systematic search with supplementary searches for existing programme theories of mental health discharge planning. This will involve extraction of data, initially from key papers, on contexts relevant to discharge planning, the outcomes relating to these contexts and the mechanisms underlying the observed relationships between context and outcome. These will be formulated into 'IF, THEN, BECAUSE' statements, for example, IF discharge is planned with involvement from service user, THEN the person if less likely to relapse postdischarge, BECAUSE the discharge meets the needs of the service user. Numerous theories are likely to be identified; therefore, the credibility and relevance to the scope of the review will be regularly assessed by the research team and LEAG members to retain the focus to the system of interest (ie, mental health inpatient discharge planning). Programme theories will be mapped against the levels of the system to ensure sufficient spread and attention to factors relevant for subsequent stages of the study.
2. Theory testing and refinement: the core review team will test and refine the IF, THEN, BECAUSE statements iteratively against findings from additional research papers, this will include discussions with the wider research team and LEAG. There are inherent biases in the literature and the lived experience perspective will help ensure that theories are relevant to those accessing services.
3. Analysis and synthesis: the engineering-based systems approach will provide a framework for analysis to ensure; programme theories, articulated as context–mechanism–outcome configurations (CMOCs), map across macro, meso and micro levels of the system. An example of 'efficiency' is illustrated in figure 2 for how

the concept might operate across the different system levels. Other factors might include risk management or social/clinical narrative about specific diagnoses. NVivo will be used to organise and track analysis. Tabulation and narrative write-up of evidence related to each programme theory will be shared with the research team and LEAG to support transparency and rigour in the analysis process. For more detail, see realist review protocol on PROSPERO.

#### Realist evaluation

We will conduct service user, carer and staff interviews, focus groups[32] and ward observations to refine the programme theories. An embedded case study design[33] will test programme theory components across systems within and between sites. Findings will be compared across the sites and participants to identify similarities and differences related to the CMOCs. We will look specifically to see how differences are linked to the contextual features of the sites and characteristics of the participants to understand how this affects the behaviour of mechanisms (ie, in which circumstances are they triggered or not and with what outcome). This is a key to ensuring transferability and acceptability of the new approach.

The interviews and focus groups will serve two purposes. First, service users and carers will be asked about their experiences prior to, during and post the inpatient admission that relate to discharge planning. Staff will be asked to explain their role in the trust, any processes, resources or strategies they use and their experiences of discharge planning with service users. This will establish the personal and professional context of participants and allow for new concepts to be identified. Second, the interviews and focus groups will take the form of a 'teacher–learner' cycle,[34] inviting participants to confirm, refute or expand components of the programme theories based on their experience. Participant deliberations of the programme theories will be contrast against the original assumptions to identify where there are disagreements and alternative explanations. This will strengthen understanding of how the context in which discharge planning takes place impacts service user and staff experience, thereby elucidating the circumstances in which mechanisms are triggered. Staff whose roles directly or indirectly impact discharge will be recruited across different levels of the system. Relevant sections of medical notes of service users recruited to interviews and focus groups (who consent to this) will be reviewed to further understand the discharge process. This includes data on route of admission, route of discharge and any documentation of discharge planning and the discharge meeting. Anonymised data will be collected to provide aggregated service user characteristics and map the discharge process.

Ward observations be conducted at each of the participating wards and will include observations in communal areas and of discharge conversations, relevant meetings and ward rounds.[35–37] They aim to support understanding of interpersonal nuance, the way that contextual factors

| System level | | Context | Mechanism | Outcome |
|---|---|---|---|---|
| **MACRO** | Public policy | Prioritisation of efficiency in public sector services | Public accountability and responsibility to the taxpayer | Policy that operationalises emphasis on efficiency |
| | Community | Efficiency measures set and monitored by government and public bodies such as the Care Quality Commission | Need to operationalise and quantify efficiency for demonstration to taxpayer | National targets set to demonstrate efficiency quantitatively – e.g. waiting times |
| | Organisational | Local policy and protocols to meet national and local targets and priorities, capture and demonstrate this | Need to demonstrate efficiency to inspection bodies and commissioners | Local targets, documentation and data capture to demonstrate efficient use of resources. |
| **MESO** | Interpersonal | Efficiency prioritised across social organisational hierarchy – executive to managers, mangers to frontline staff, registered staff to non-registered staff, ultimately frontline staff to service users | Pressure to demonstrate efficient use of resources to senior members of staff | Care can become 'task-focussed' rather then person centred. Discharge choices prioritise system need (releasing beds) rather than service user need |
| **MICRO** | Individual | Limited resources to meet service user need<br><br>Staff internalise pressure to deliver efficient care (meet the needs of the system)<br><br>Care does not meet service user's mental health needs | Service user: aware of staff pressure or subject to stresses in ward environment<br><br>Staff: pressure to demonstrate efficacy causes stress and moral distress as it conflicts with desire to deliver person-centred care | Discharge planning is rushed, non-collaborative and does not meet service user's needs<br><br>Staff experience high rates of stress, moral distress, and burn-out. This limits ability to deliver compassionate person-centred care |

DIRECTION OF INFLUENCE

Point of discharge planning delivery

**Figure 2** Example of efficiency as a context mechanism outcome configuration across macro, meso and micro levels of the system.

relate to outcomes and insights into causal mechanisms. Data will be collected using a template reflecting the programme theories. We will conduct a review of policy and strategy documents to provide an account of stated organisational aims and priorities for discharge and how documents are structured to support the process.

## Analysis
The analysis will follow Realist and Meta-narrative Evidence Syntheses: Evolving Standards quality standards,[38] using realist logic.[20 21] A core team will work with LEAG members to iteratively evaluate data in relation to the programme theories to facilitate theory refinement. NVivo[39] will support data management and analysis. Data coding will be deductive (informed by our initial programme theories), inductive (derived from the collected data) and reproductive (making inferences about mechanisms based on interpretations of our data to infer underlying causal processes). Evidence tables will be produced to demonstrate theory refinement.

## Outputs
The outputs for stage 1 will include an evidence-based programme theories of discharge preparation and planning, a rich understanding of context, including the stakeholders involved and their wants and needs from the discharge process and a set of causal mechanisms operating within the discharge contexts.

## Stage 2: Codesign a systemic discharge intervention
### Aim
Informed by the programme theories, codesign a sustainable systemic discharge intervention that meets service user needs, is compatible with how staff work, and feasible to implement.

### Objectives
Use a healthcare engineering-based systems approach as a framework to develop:
1. An agreed scope for the factors that can be changed within the discharge planning approach.
2. A systemic discharge solution that has the potential to balance key wants and needs of all stakeholders.

3. Methods for measuring the performance of the proposed solution against key wants and needs of service users and other stakeholders.

### Design
Stage 2 uses the Engineering Better Care framework and Improving Improvement toolkit (IItoolkit, www.iitoolkit.com).[40] This is a systems-based engineering approach that aligns with complex intervention development, as it is non-linear, creative and forward looking to future evaluation. The programme theories from stage 1 will provide an understanding of the context and definition of the problem across the wider system of interest (critical stages for the Engineering Better Care approach prior to designing the solution).

### Prioritisation workshop
The wants and needs service users and other key stakeholders may conflict. In this case, the research team and LEAG will review these, in combination with the agreed scope. The following structure (MoSCoW method)[41] will be used. This will categorise wants and needs into 'must haves' (core essential needs for an improved discharge process), 'should haves' (highest priority 'wants'), 'could haves' (secondary priority 'wants') and 'won't haves' (rejected as being incompatible with the agreed scope).

The prioritised wants and needs embody what a 'better' solution would mean, across the perspectives of stakeholders, while acknowledging the pragmatic reality of delivery and resource limitations.

### Exploratory design workshops
Iterative 3 hour exploratory design workshops will be conducted at each study site with service users, staff and carers/supporters. The research team will use the IItoolkit to develop ideas and proposals for an improved discharge process to meet the discharge needs of service users identified from the realist review and evaluation. The tools and activities within the IItoolkit will be used to support an iterative process of problem-finding and problem-solving, in a systems context. This encourages divergent thinking to stimulate ideas about how the discharge process can be improved, and convergent thinking to consider how these ideas can be selected, refined and developed to produce a small set of feasible concepts. This will challenge the understanding and insights gathered from the realist evaluation and the scope of what can be delivered.

### Review and refinement workshop
The exploratory design workshop outputs will be considered at two 3-hour sessions with the research team and LEAG. The research team and LEAG will review, refine and evaluate the ideas and concepts from the exploratory workshops, to give a recommended lead proposal for an improved discharge process. This may involve developing tools and/or materials to better assist discharge planning and/or reconfiguring the discharge process and/or updating the guidance for the discharge process. Examples might include a combination of training materials, clinical supervision or reflective practice templates, a discharge planning group outline or collaborative discharge planning tools or documentation. Team members involved in this work contribute skills in systems engineering, risk assessment and design, psychology, nursing, psychiatry and lived experience of discharge.

### Feedback sessions
The new discharge intervention will be reviewed and refined during feedback sessions, with service users, carers/supporters and clinical staff from the design workshops. This will focus on acceptability and implimentability of the new approach. Assessment will be based on the prioritised wants and needs that informed the design of the new intervention, together with the success measures. Staff will be asked to develop an implementation plan with the research team to support use of the new discharge approach on their ward. These plans will be taken to additional meetings with staff on the research wards to agree plans for implementation of the new approach on their wards.

### Explanatory model
The research team and LEAG members will agree how the proposed solution could be practically measured against agreed wants and needs. The programme theories, prioritised discharge needs and identified outcomes associated with these and the tools and/or materials for the new systemic discharge intervention will be used by the research team and LEAG to develop a realist-informed explanatory model, including resources, activities and measurable process indicators for implementation. This will include the components of the discharge intervention to be implemented, steps to implementation, process indicators of successful implementation, measures of acceptability, cost impact and outcomes of effective discharge. This will support data collection for stage 3.

### Outputs
The outputs for stage 2 will include prioritised wants and needs for an improved discharge process, practical measures of success that are aligned with these prioritised wants and needs, the as an improved discharge process, and an explanatory model to support implementation and evaluation.

### Stage 3: Evaluate acceptability, implementation and cost–impact of the new discharge intervention
#### Aims
1. Evaluate acceptability and implementation of the discharge intervention.
2. Explore resource and cost implications and determine feasibility of collecting data for a future economic evaluation.
3. Inform a final specification for the discharge intervention that can be tested in a future Hybrid Type II trial that will determine its effectiveness and impact, including economics.

## Objectives

1. Understand acceptability, barriers and facilitators to implement SDCA.
2. Evaluate how delivery and fidelity is shaped by the healthcare context.
3. Measure reach, adoption and maintenance.
4. Risk-assess the use of the intervention.
5. Estimate resource and associated costs impact.
6. Identify recommendations for optimisation, wider implementation and future evaluation.
7. Evaluate feasibility of collecting service user outcome and economic evaluation data.

## Design

The explanatory model will support implementation and a process evaluation for stage 3. A parallel, mixed methods process evaluation will assess the feasibility of implementation, acceptability, risks and benefits and cost impact of the discharge intervention.

## Ward-based observations

Ward-based observations will be conducted to investigate implementation of the discharge intervention and how this interacts with the ward and wider contexts. This will include observing discharge planning consultations and system strengthening components (eg, training) as well as ward-based processes that impact on delivery. Researchers will attend relevant meetings (eg, reviews and discharge planning meetings), observe training sessions or other relevant interactions and collect data in the form of field notes. This will evaluate whether the components of the discharge approach have fidelity in terms of what was designed in the workshops, and whether it impacts the areas of service user discharge need identified and prioritised from the programme theories.

## Semistructured interviews with service users, carers/supporters, and staff

Interviews with service users will gain perspectives on the acceptability of the discharge intervention, with a specific focus on how the resulting discharge plan supported their transition from the ward to home, the quality of collaboration between themselves and ward staff and whether their discharge plans were supportive of a safe and effective transition from the ward to home. This will be informed by the programme theories and prioritised needs. Interviews with staff will be carried out 6 months after commencing use of the discharge intervention to allow it to be embedded into routine practice, obtaining perspectives on acceptability, barriers and facilitators to implementation, impact on quality of care over time and recommendations for widescale implementation. Interviews will be semistructured with topic guides informed by the prioritised discharge needs of service users and possible barriers and facilitators in terms of implementation.

## Service user and outcome data

The feasibility of collecting service user demographics and outcome data from routine medical records and questionnaires will be assessed, including readmission, suicidality, mental health symptoms, personal recovery and quality of life. Participants will complete a questionnaire containing selected measures and resource use questions to inform data collection feasibility for future evaluation, resource use and associated cost analysis. This will include the outcome measures identified from the programme theories and a resource use questionnaire. Medical records will be reviewed to assess the reach, adoption and maintenance of the discharge intervention.

## Data analysis

### Qualitative analysis

Fieldnotes from the ward-based observations, the data from document reviews and interviews with staff and service users will be compared with the context–mechanism–outcome configurations identified in stage 1 and the realist-informed explanatory model to explore whether changes to practice occurred and met the prioritised needs as theorised. This will follow the realist logic of analysis used in stage 1. The observations and document review data will be used as measures of process indication for implementation identified in the explanatory model. This will indicate fidelity to implementation. We will also use descriptive analysis of the data from the document reviews to describe reach and adoptions (ie, the extent of use and who it is being used with). The interviews evaluate how the process and content of the discharge intervention 'worked' from the participants' perspective, aiming to understand the quality of collaboration, usefulness of the discharge intervention and barriers and facilitators to implementation. A constant comparison approach will be adopted, working iteratively between data obtained from different interviewees within and between wards and case study sites. We will also analyse how different intervention components interact with relevant macro (eg, national policy); meso (eg, in-patient ward protocols, staff arrangements, other services) and micro (eg, communication and behaviour within discharge planning encounters) contextual features relevant to scaled-up implementation. This will be undertaken with support of the LEAG.

### Quantitative analysis

Statistical analysis will include descriptive analyses of changes over time (eg, numbers of discharge plans) and graphical plotting of changes, comparing trends between wards, both descriptively and potentially with regression. Additional analyses prompted by qualitative findings (eg, effects of the discharge intervention on specific groups or diagnoses) will be explored. Completion rates and patterns of data collection tools will be descriptively analysed to inform the data collection feasibility for future trialling.

## Resource use and costing analysis

Recorded resource use will be multiplied by standard unit costs.[42] A key costing perspective will be that of the NHS and Social Services, but we will also disaggregate costs to consider those incurred by (1) the inpatient wards; (2) other providers and (3) service users (eg, out of pocket costs). This will consider which costs are one-off (eg, training) and recurring across levels of the system. Return rates and levels/patterns of missing data on the resource use questions will be descriptively analysed to inform the feasibility of a future economic evaluation and refinements to the questionnaires to improve completion rates. Extraction of related data from routine sources will also be explored to further inform future evaluation.

## Stakeholder focus groups

Two stakeholder focus groups (6–8 participants per group, 12–16 in total) will be carried out towards the end of stage 3 to identify how to optimise the discharge intervention for wide-scale implementation and to determine priorities for a future trial. The key stakeholders will include a mixture of key stakeholders including mental health staff, service directors and policymakers who can provide critical insight into wider implementation. We will share findings and ask stakeholders to make recommendations for finalising the design and content of the SDCA and required system strengthening components to optimise intervention implementation. We will map components against the implementation strategies identified by the Expert Recommendations for Implementing Change[43] to finalise the SDCA.

## Stage 3: outputs

The outputs for stage 3 will include a finalised systemic discharge intervention ready for implementation and trialling, refined programme theories setting out the factors necessary for implementation, estimation of the cost and resource impact, initial feasibility data for a hybrid type 2 trial of the intervention, including identified service user outcomes, process/implementation indicators and economic measures.

## Recruitment and consent

We are recruiting and consenting five groups of service users, carers/supporters and staff (see figure 1):
1. Service users who are currently admitted for the ward-based observations (stages 1 and 3).
2. Service users in the community who have had experience of discharge from a mental health ward within the last 12 months for interviews and focus groups (stages 1 and 2).
3. Service users who are being discharged (stage 3).
4. Carers/supporters of people who have been discharged in the last 12 months (stages 1, 2 and 3).
5. Staff who have roles that impact on inpatient discharge (stages 1, 2 and 3).

Recruitment of service users, carers/supporters and staff for interviews, surveys and groups will be purposive. The LEAG will advise on ways to maximise access and participation—including groups that may be at risk of under-representation due to diagnosis, ethnic background or other demographic factors. Potential participants will be approached via multiple channels to increase access and participation, including, via clinical teams, through participation channels, and promotion including posters on wards and other service-user facing clinical spaces.

For interviews, focus groups and workshops, a research team member will arrange a consent meeting at least 48 hours after receipt of the participant information sheet (PIS). It will be established that the participant has read this, understands the study and implications of participation and any questions are answered. Capacity to consent will be assessed.

On the days of the ward observation, information posters will be displayed in areas where the observations are taking place. All staff and service users will be given verbal information and a simplified PIS about the reason for the observations and be asked to verbally consent to the observations. This simplified consent process has been designed to minimise burden and confusion for service users. Observers will be wearing a lanyard that makes it clear who they are and that they are undertaking observations. If approached, they will answer any questions transparently. Service users will be informed that they can choose to opt out of the observations at any time (they are also free to leave the observed space). Staff will be asked to opt out if they do not want to be observed. Staff will be informed that if they are concerned about observations, including a particular service user, or if they become concerned about anybody during the observation, they can ask for the observation to be moved or terminated.

All service-users who are discharged from the 3 study sites within the first 6 months of stage 3 will be asked whether they wish to opt out of their routine data being used for research purposes.[44–46]

## ETHICS AND DISSEMINATION

MINDS includes protocols for managing distress or safety issues relating to interviews, focus groups and ward observations, which have received ethical approval for stage 1. These will also be applied to the activity for stages 2 and 3. The SSC and LEAG will support ethical issues encountered during the study.

## Dissemination

We will work with the LEAG, to develop open access peer-reviewed journal publications and conference presentations. We will establish routes to engage with public and service user communities, including blogs, podcasts and videos via partner Mind and reaching out to other organisations, for example, National Survivor User Network (NSUN). This is NIHR Applied Research Collaboration (ARC) East of England affiliated project and findings will

be disseminated in an assessable form via ARC platforms and networks.

**Author affiliations**
[1]Research and Development, Norfolk and Suffolk NHS Foundation Trust, Norwich, UK
[2]Noriwch Medical School, The University of East Anglia, Norwich, UK
[3]Department of Enginering, University of Cambridge School of Technology, Cambridge, UK
[4]Centre for Research in Public Health and Community Care, University of Hertfordshire, Hatfield, UK
[5]School of Life Course and Population Sciences, King's College London, London, UK
[6]NIHR Collaboration for Leadership in Applied Health Research & Care (CLAHRC) East of England, Cambridge, UK
[7]Norwich Medical School, University of East Anglia, Norwich, UK
[8]Engineering Design Centre, University of Cambridge, Cambridge, UK
[9]School of Psychology and Sports Science, Anglia Ruskin University, Chelmsford, UK
[10]London College of Fashion, University of the Arts London, London, UK
[11]Centre for Research in Primary & Community Care, University of Hertfordshire, Hatfield, UK
[12]Independant, Cambridge, UK
[13]Norfolk and Suffolk NHS Foundation Trust, Norwich, UK
[14]Health Services and Population Research, King's College London, London, UK
[15]Department of Social Work, Education & Community Wellbeing, Faculty of Health and Life Sciences, Northumbria University, Newcastle upon Tyne, UK
[16]Hertfordshire and Peterborough NHS Foundation Trust, St Albans, UK
[17]Department of Engineering, University of Cambridge, Cambridge, UK
[18]Wolfson Institute of Population Health, Queen Mary University of London, London, UK
[19]East London NHS Foundation Trust, London, UK

**Acknowledgements** We would like to thank LEAG members including Hajara Begum, Katie Fillingham, June Hanshaw, John Lucas, Isaac Samuels, and Roger Talbot for their ongoing input across the MINDS project; helping to ensure the processes and outputs reflect the experiences and needs of people with lived experience and their supporters.

**Contributors** CHac took the lead role in design, coordinating the design and writing the protocol. MH, EK, JJ, HZ, SMD and CHai led the design of stage 1. AK, CHac, SW, SB, TK and JC led the design of stage 2. JM and APW led the design of stage 3. SR, JW and LG contributed to the design across the project and the writing and editing of the protocol. SR, HZ and SB led on the lived-experience input to the conception and design across the project. JB, AS, JW, FR, GB, ZT and CHac led on clinical perspectives, clinical governance, distress management and staff input across the design of the protocol. All authors were involved in editing the protocol.

**Funding** This work is funded by the NIHR (NIHR133013). APW is supported by the NIHR ARC EoE. The views expressed are those of the authors and not necessarily those of the NIHR or the Department of Health and Social Care.

**Competing interests** None declared.

**Patient and public involvement** Patients and/or the public were involved in the design, or conduct, or reporting, or dissemination plans of this research. Refer to the Methods section for further details.

**Patient consent for publication** Not applicable.

**Provenance and peer review** Not commissioned; externally peer reviewed.

**ORCID iDs**
Corinna Hackmann http://orcid.org/0000-0002-4940-6998
Adam P Wagner http://orcid.org/0000-0002-9101-3477
Lisa Marie Grünwald http://orcid.org/0000-0001-7010-871X
Sonia Michelle Dalkin http://orcid.org/0000-0002-3266-5926
Timoleon Kipouros http://orcid.org/0000-0003-3392-283X

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
