## [Reviewer comments · BMJ Open]

ARTICLE DETAILS

TITLE (PROVISIONAL)	Co-designing a systemic discharge intervention for inpatient mental health settings (MINDS): a protocol for integrating realist evaluation and an engineering-based systems approach
AUTHORS	Hackmann, Corinna; Komashie, Alexander; Handley, Melanie; Murdoch, Jamie; Wagner, Adam; Grünwald, Lisa Marie; Waller, Sam; Kaminskiy, Emma; Zeilig, Hannah; Jones, Julia; Bray, Joy; Bagge, Sophie; Simpson, Alan; Dalkin, Sonia; Clarkson, John; Borghini, Giovanni; Kipouros, Timoleon; Rohricht, Frank; Taousi, Zohra; Haighton, Catherine; Rae, Sarah; Wilson, Jon

VERSION 1 – REVIEW

REVIEWER	Weich, Scott University of Sheffield, School of Health and Related Research
REVIEW RETURNED	25-Jan-2023

GENERAL COMMENTS	This is a well written protocol paper describing an important study. It is suitable for publication and will be of considerable interest. I wonder, however, whether the presentation might be improved, specifically with the use of (far) fewer bullet points. I would also like to see a Discussion section describing the strengths and limitations of the proposed design. I am mindful that this is a non-randomised, unblinded study and inferring causality is going to be (potentially) tricky. That said, I think the authors could add to the literature by making a strong case for their methodology, which has a lot to recommend it. It is important to add to the debate about why RCTs are not the best way to evaluate complex, systems-aware and context-dependent interventions. I am also interested in how the findings will be translated into policy and practice, and these sections could be expanded.
---

REVIEWER	Durbin, Janet Center for Addiction and Mental Health
REVIEW RETURNED	13-Feb-2023

GENERAL COMMENTS	This paper outlines the protocol for a study to develop and test a discharge protocol from inpatient mental health to community care. The compelling rationale stated in the introduction is the high risk of adverse outcomes in the months following discharge and the lack of guidance on how to improve practice. The paper emphasizes that individuals with lived experience and other stakeholders will work in a co-production arrangement to lead and inform all phases of the work. The study is addressing an important need and has suggested some novel strategies to do so. However I found the paper very hard to follow.
---

	For example, re WP1: This component will first include a review of program theories of discharge planning (what is an example of such a theory) which will then be compared against 'evidence'. What evidence is this - the review of service user, carer and staff experiences of discharge planning mentioned earlier in the section? Lived experience of study participants will also be included. NVivo will be used to tabulate and write up the analysis related to each theory but I am unclear what the aim is – is it to show how experiences reported in the literature align with each of the theories? I would need an example to understand this. Also how many theories do you expect to find/ expect to review (how will you decide). Then interviews, focus groups, ward observation will be conducted to refine the theories. How so? Medical notes will be reviewed and anonymized data will be collected. Again how so? Case studies will be developed to test the theory components across systems and within and between sites. Again I have difficulty understanding how this will be accomplished and what the case studies will include. It might help the reader (and be more concrete) to describe one possible transition theory and then follow it through to show how the subsequent analyses will be used to assess it. Also more detail on each of the data collection methods and purpose is needed. For example, what do the researchers expect to learn from ward observation, and how will they do it? What questions will guide the interviews and focus groups. How will medical notes and anonymized data be accessed and to what purpose. It is not clear what realist evaluation and the engineering approach are specifically contributing to the process. I am not reviewing the other components but similar concerns are present. Additionally, re WP3, there is no information on how the SDCA will be implemented. Implementation of a new practice is a significant undertaking. Many acronyms are used – again challenging the reader. Can these be reduced. For example, can WP1 be changed to phase 1 or stage 1. I did not see any study timelines. This looks like important innovative work and the co-production approach is a strength. However it needs a much clearer presentation.
--	---

VERSION 1 – AUTHOR RESPONSE

Reviewer: 1

Prof. Scott Weich, University of Sheffield Comments to the Author:

'This is a well written protocol paper describing an important study. It is suitable for publication and will be of considerable interest.

I wonder, however, whether the presentation might be improved, specifically with the use of (far) fewer bullet points.'

Our response: Bullet points have been removed throughout the protocol

'I would also like to see a Discussion section describing the strengths and limitations of the proposed design. I am mindful that this is a non-randomised, unblinded study and inferring causality is going to be (potentially) tricky. That said, I think the authors could add to the literature by making a strong case

for their methodology, which has a lot to recommend it. It is important to add to the debate about why RCTs are not the best way to evaluate complex, systems-aware and context-dependent interventions.'

Our response: Thank you for this suggestion. We were unable to add a discussion section to the protocol. We have added more detail on why the proposed design was selected in the 'study design' section. The protocol is for intervention development and aligns with MRC complex intervention guidelines. We do intend to conduct a Hybrid type II clinical trial in a future study to test the clinical effectiveness of the intervention that is designed in the MINDS study. We have attempted to make this clearer in the protocol.

'I am also interested in how the findings will be translated into policy and practice, and these sections could be expanded.'

Our response: The aim is for the MINDS study is to co-design the discharge intervention. This will be clinically trialled in another, future, study. We have removed the sections from the 'dissemination' section about our engagement with NICE to discuss adoption of the outputs of this wider programme of work into NICE guidelines. This is because this will be integrated into the clinical trialling, rather than the intervention development work. This is also the case for other plans for translation into policy and practice.

Editors note- Please note discussions are not part of the journal format for Protocols

Reviewer: 2

Dr. Janet Durbin, Center for Addiction and Mental Health Comments to the Author:

'This paper outlines the protocol for a study to develop and test a discharge protocol from inpatient mental health to community care. The compelling rationale stated in the introduction is the high risk of adverse outcomes in the months following discharge and the lack of guidance on how to improve practice. The paper emphasizes that individuals with lived experience and other stakeholders will work in a co-production arrangement to lead and inform all phases of the work. The study is addressing an important need and has suggested some novel strategies to do so. However I found the paper very hard to follow.

For example, re WP1: This component will first include a review of program theories of discharge planning (what is an example of such a theory) which will then be compared against 'evidence'.'

Our response: Thank you for your detailed review of the protocol paper. We have included examples of this both in the second paragraph (phase 1) of the 'realist review' section (narrative example) and in the fourth paragraph (phase 3) of the 'realist review' section (Figure 2)

'What evidence is this - the review of service user, carer and staff experiences of discharge planning mentioned earlier in the section ?'

Our response: We have clarified this in the realist review section in the third and fourth paragraph (phases 2 and 3 of the realist review)

'Lived experience of study participants will also be included. NVivo will be used to tabulate and write up the analysis related to each theory but I am unclear what the aim is – is it to show how experiences reported in the literature align with each of the theories? I would need an example to understand this.'

Our response: An example has been included in the second paragraph (phase 1) of the Realist Review section. This demonstrates how the data will be extracted from the literature and developed into theories.

'Also how many theories do you expect to find/ expect to review (how will you decide).'

Our response: It is hard to estimate the number of theories we anticipate that we will find. We have included a discussion in the second paragraph (phase 1) of the realist review section about how the credibility and relevance to the scope of the review will be regularly assessed to ensure we maintain focus on the system of interest (i.e., mental health inpatient discharge planning)

'Then interviews, focus groups, ward observation will be conducted to refine the theories. How so?'

Our response: We have added more detail and clarity on this in the second and third paragraphs of the realist evaluation section.

'Medical notes will be reviewed and anonymized data will be collected. Again how so?'

Our response: We have added more details on the review of medical notes in the second paragraph of the realist evaluation section.

'Case studies will be developed to test the theory components across systems and within and between sites. Again I have difficulty understanding how this will be accomplished and what the case studies will include.'

Our response: More detail has been added regarding the case site analysis in the first paragraph of the realist evaluation section.

'It might help the reader (and be more concrete) to describe one possible transition theory and then follow it through to show how the subsequent analyses will be used to assess it.'

Our response: Prior to analysis, we are unable to describe a theory being developed through the processes of analysis in any detail. We have included examples in both the 'realist review' section narrative and as a figure (Figure 2) to enable the reader to visualise how the theories might develop from the various data sources.

'Also more detail on each of the data collection methods and purpose is needed. For example, what do the researchers expect to learn from ward observation, and how will they do it?'

Our response: More detail on the data collection methods and purpose have been added to the second and third paragraphs of the 'realist evaluation' section.

'What questions will guide the interviews and focus groups.'

Our response: We have added more details on the questions for interviews and focus groups in paragraph 2 of the 'realist evaluation' section.

How will medical notes and anonymized data be accessed and to what purpose.

Our response: More detail has been added to paragraph 2 of the realist evaluation section to address that the medical notes of participants who consent to this will be accessed – and the purpose of this.

It is not clear what realist evaluation and the engineering approach are specifically contributing to the process.'

Our response: We have clarified this throughout the protocol. Specifically, we have addressed this in the 'analysis and synthesis' section (phase 3) of the 'realist review' section. We have also included visual example (Figure2) to help the reader visualise the realist evaluation and engineering approaches are integrated for theory building.

'I am not reviewing the other components but similar concerns are present.'

Our response: We have added more detail across the different stages of the project to clarify data collection methods and how these will translate into the research and intervention development outputs.

' Additionally, re WP3, there is no information on how the SDCA will be implemented. Implementation of a new practice is a significant undertaking.'

Our response: We have added more detail on how the implementation of the discharge intervention will be supported in the 'feedback sessions' and 'explanatory model' sections towards the end of stage 2.

'Many acronyms are used – again challenging the reader . Can these be reduced. For example, can WP1 be changed to phase 1 or stage 1. I did not see any study timelines.'

Our response: All acronyms have been removed where possible – including changing WPs (work packages) to stages.

'This looks like important innovative work and the co-production approach is a strength. However it needs a much clearer presentation.'

Our response: Thank you. We have improved the presentation throughout to respond to your detailed review comments.

VERSION 2 – REVIEW

REVIEWER	Weich, Scott University of Sheffield, School of Health and Related Research
REVIEW RETURNED	03-Apr-2023

GENERAL COMMENTS	Thank you for sharing the revised manuscript, and for the responses to reviewers' comments. The paper has been improved, and all of my comments addressed.
--

REVIEWER	Durbin, Janet Center for Addiction and Mental Health
REVIEW RETURNED	10-Apr-2023

GENERAL COMMENTS	Thank you for these extensive revisions in response to reviewer feedback. I found the paper to be much clearer and the proposed study to be creative, with many strengths. One request is that you clarify in the text the difference between case study ward and
---

	research site award. This may be an issue of sample size but clarification would be helpful.
--	--

VERSION 2 – AUTHOR RESPONSE

Reviewer: 1

Prof. Scott Weich, University of Sheffield Comments to the Author:

Thank you for sharing the revised manuscript, and for the responses to reviewers' comments. The paper has been improved, and all of my comments addressed.

Our response: Thank you for your review and comment.

Reviewer: 2

Dr. Janet Durbin, Center for Addiction and Mental Health Comments to the Author:

Thank you for these extensive revisions in response to reviewer feedback. I found the paper to be much clearer and the proposed study to be creative, with many strengths. One request is that you clarify in the text the difference between case study ward and research site award. This may be an issue of sample size but clarification would be helpful.

Our response: Thank you for your review and feedback. There is no difference between a case study ward and research site ward (case study site and research site were used interchangeably). We have removed all references to research site to avoid confusion and clarify this.